# The Crossroads of Ecotourism Dependency, Food Security and a Global Pandemic in Galápagos, Ecuador

**Adam Burke**

Department of History, Humanities & International Studies, Hawaii Pacific University, 1 Aloha Drive, Honolulu, HI 96813, USA; aburke@hpu.edu

**Abstract:** International esteem for Galápagos' natural wonders and the democratization of travel have contributed to a 300% increase in annual tourist entries to the archipelago from 2000 (68,989) to 2018 (275,817). The attendant spike in tourism-related anthropogenic impact coupled with deficient infrastructure development has put the archipelago's natural capital and carrying capacity at risk. The complex nature of Galápagos' food insecurity is linked to the archipelago's geographic isolation, its diminishing agricultural workforce, international tourists' demand for recognizable food, and a lack of investment in sustainable and innovative agricultural futures. Food security is key to the long-term well-being of Galapagueños, who sustain Galápagos' tourism industry. However, the COVID-19 pandemic has further exposed the vulnerability of human systems in Galápagos, especially the fragility of Galápagos' ecotourism dependency. Galapagueños' struggle to endure the tourism sector's slow rebound following the 2020 travel restrictions points to an urgent need to implement food security measures as an indispensable component of the archipelago's long-term sustainability plan. This article presents ethnographic data to discuss the tourism sector's impact on local food systems, Galapagueños' right to food sovereignty, efforts to increase agricultural production, and why strengthening institutional partnerships is vital to Galápagos' food self-sufficiency.

**Keywords:** Galapagos; tourism; ecotourism; sustainability; food security; food self-sufficiency; food sovereignty; complexity





## 1. Introduction

Tourism has replaced agriculture and artisanal fishing as Galápagos' leading economic sector. Before the 2008 global recession, the archipelago's tourism services accounted for 71% of the "gross island product" [1]. This statistic far surpasses Wood's (2017) assessment that island states are on the road to tourism dependency once tourism accounts for more than 25% of the export economy [2] (p. 96). After recovery from the 2008 global recession, Galápagos' tourism industry increased 5.1% annually between 2010–2019—with tourism entries to the Galápagos National Park (GNP) increasing from 173,297 to 271,238 during that decade [3–5]. This growth rate exceeds the World Tourism Organization's (WTO) pre-COVID projection for the global tourism industry to increase 3.3% annually from 2018 to 2030 [6]. Galápagos' ecotourism boom has led to a state of "over tourism" and burdened Galapagueños to resolve fundamental issues of socioeconomic and ecological well-being.

To this point, Galápagos' 30,000 permanent residents and non-resident local stakeholders are indeed partnering to address post-COVID realities, especially the tourism sector's role in and impact on social and natural systems. Their attention considers a problematic set of socioeconomic and ecocultural issues, including but not limited to overutilized and deficient infrastructure (e.g., potable, wastewater systems), solid waste management, modernizing (renewable) energy systems, climate change, extractive industrial fishing, common pool resource management, the continuity of traditional vocations (e.g., fishing, farming), and the underdevelopment of social services (e.g., health care, education). My prior research suggests that these challenges are compounded by perceptions that

exogenous economic and conservationist interests in Galápagos have "(i) embedded a binary land–sea disposition, (ii) established residents' dependency on ecotourism as a dominant economic sector, (iii) crept into local politics and reaped financial benefit via economic leakage and social stratification, and (iv) made permanent residents dependent upon marine ecotourism yet dissociated from the sea" [5] (p. 64).

Other archipelagos and Small Island Developing States are similarly dealing with developing tourism sustainably. For example, stakeholders from across the Pacific gathered at an April 2020 Hawaii Green Growth virtual conference to discuss how islands are experiencing and addressing COVID-19-related disruptions to the tourism economy along with resiliency strategies to "bounce back better". At that forum, then Galápagos governor Norman Wray explained that tourism development initiatives overshadow Galápagos' glaring vulnerability to food insecurity. Wray remarked that food security is key to the Galápagos tourism sector's long-term viability. This study understands that food security is achieved "when all people, at all times, have physical and economic access to sufficient, safe and nutritious food to meet their dietary needs and food preferences for an active and healthy life" [7]. However, what is the current state of food security in Galápagos, and what questions should be addressed when planning for the archipelago's sustainable future?

For perspective, Sampedro et al. (2018) reported that "approximately 75% of the agricultural food supply was transported from the mainland in 2017" and that this number "will increase to 95% by 2037 with no changes in food policy" [8] (p. 1). The authors advise that any attempt to increase tourist arrivals must also incorporate plans to promote local agricultural capacities. Synergizing tourism and food production initiatives empowers a development strategy that supports Galapagueños' well-being and subsistence needs [8]. However, how and to what extent should Galápagos stakeholders rely upon food imports in the short- and long-term? What are the tourism externalities that complicate food security initiatives? How do local stakeholders understand and envision food security as a driver of the tourism industry's long-term sustainability? As an entry point to addressing these questions, the following section first unpacks how notions of risk feature in Galápagos histories and why risk is an essential conceptual framing to understand Galápagos' eco-political landscapes and seascapes.

## 2. Galápagos "at Risk"

Economic crises (e.g., the 2008 global financial crisis, the 2014–2016 global collapse in oil prices, the 2020 tourism shutdown) have exposed the vulnerability of Galápagos' tourism economy and dependency on global markets. Galápagos' ongoing response to the COVID-19-related travel bans and tourism restrictions has renewed risk assessment discussions. This section briefly outlines how risk perceptions in Galápagos have changed over time and why understanding notions of risk are critical to analyzing the nexus between tourism development and food security in Galápagos. It does so by introducing (i) the risk mitigation of early settlers prior to the archipelago's development as a tourism hotspot, (ii) how risk features in the archipelago's conservationism and sustainable development ideologies, (iii) the global tourism project and its attendant vulnerabilities, (iv) the risks associated with sustainable tourism development in Galápagos, and (v) how these factors resonate with the precarity of the archipelago's food systems and diminishing agricultural production.

### 2.1. Risk Tolerance

The tripartite project of conservation development, ecological stewardship, and resiliency building is relatively modern in Galápagos. The majority of Galápagos' human occupation reflects an extractive "human-in-nature" relationship [9,10]. Accounts of Galápagos' early settlement attempts detail the perilous and celebrated journeys of those who pioneered Galápagos' first communities, e.g., [11–16]. Early colonial stations (e.g., Villamil in 1832, Cobos in 1879) endured Galápagos' reputation as a "fiery wasteland" [16].



Human settlement in Galápagos remained perilous up until the mid-20th century due to the archipelago's provisional infrastructure and isolation from the exterior.

Nonetheless, the Enchanted Isles' allure attracted waves of migrants in the latter 20th century. For instance, migrants such as Pepito Herrera and other continental Ecuadorians decided to risk relocating to Galápagos during WWII to assist the U.S. military's efforts to build a base on Baltra Island. Other continental Ecuadorians sought financial prosperity when migrating to work in Galápagos' emerging fishing and agricultural industries. In this light, the bedrock of Galápagos' pioneering histories—and notions of risk—were anthropocentric as settlers endured Galápagos' rugged conditions. The advent of modern conservationism in the latter 20th century introduced a "humans-with-nature" mindset, e.g., [5,17], and shifted the narrative from risk tolerance to risk assessment

### 2.2. Risk Assessment

Today, the livelihoods of Galapagueños are substantially more bearable than their colonial predecessor's thanks to advances in public health, municipal infrastructure, and communication technologies. However, residents commonly describe their ontologies and epistemologies as being "at risk" due to tourism "underdevelopment", economic leakage, and increasing labor migration [5,10,18,19]. What, then, led to the established anthropocentric focus on risk tolerance being replaced with concern over ecological precarity in the late 20th century?

Ultimately, recognizing Galápagos' ecological fragility and potential as an ecotourism hotspot contributed to founding the GNP and the Charles Darwin Research Station (CDRS) in 1959. Since their establishment, these institutions have worked together to promote the twin mantras shaping the archipelago's environmental management: "Galápagos as a natural laboratory" and "Galápagos at risk". The notion of Galápagos as a natural laboratory evolved as a process wherein visitors' (e.g., scientists, adventurers) descriptions reinforced the archipelago as a pristine ecological wonder. The crafting of Galápagos as a "natural laboratory" has also enabled the tourism industry to benefit from the commercialization of the conservation science sector's research [15,20,21].

The recognition of "Galápagos at risk" generally calls attention to the Anthropocene (i.e., a new geological era, beginning in 1950 and marked by the Great Acceleration of human consumption, which has led to recognizable ecological harm) over human precarity [22]. In the Galápagos context, this risk flags how exponential ecotourism growth is linked to projections of wide-scale species loss and ecosystem destruction [22]. The Galápagos National Park Directorate (GNPD), which oversees conservation and management of the GNP, has partnered with the CDRS for decades to reinforce these twin mantras through sustainability science research and leadership. This project has encouraged residents to replace their view of Galápagos as an inhospitable landscape requiring human domination through ranching and farming with an awareness that the archipelago's ecological vulnerability requires stewardship [9].

Accordingly, the GNPD–CDRS partnership laid the foundation for significant conservation milestones such as Galápagos' inscription as a UNESCO World Heritage site in 1978, its recognition as a Biosphere Reserve in 1984, and the creation of the 135,000 square kilometer Galápagos Marine Reserve (GMR) in 1998 [23,24]. The Ecuadorian government's further assessment of Galápagos' ecological fragility led former president Rafael to declare Galápagos "at risk" in 2007 in concordance with UNESCO's temporary listing of Galápagos as an endangered heritage site in the same year [25]. During the late 20th century, conservationist initiatives thus contributed to a gradual reconstruction of Galapagueño dispositions from survivalists to stewards of the archipelago's natural capital. This conceptual shift is noteworthy, as it reframed the human–nature trope and marks an ideological transition from risk tolerance to risk assessment.

In conjunction with conservationism development, the onset of formalized tourism in 1969 also contributed to entrenching tourists' perception of Galápagos as both a "natural laboratory" and "at risk". Global tourists' pilgrimages to Galápagos are commonly moti-

vated by a desire to experience the archipelago's pristine natural wonders before human impact causes irreplaceable harm to the ecosystems. Global tourists' fear of missing out on authentic nature experiences resonates with Laso's (2020) perspective that the collective impacts of long-term human settlement and unregulated tourism infrastructure development have made "pristine ecosystems an increasingly rare and often artificial experience" [10] (p. 138).

In this light, Galapagueños have embraced the reality that their archipelago's natural systems are at risk, which is largely due to the exponential increase in tourist numbers (see Section 2.4). However, while most Galapagueños and other social actors in the archipelago indeed endorse the "nature at risk" narrative, they disagree about "the nature of the risk and how to solve the many conflicts associated with the twin goals of economic development and resource conservation" [26,27] (p. 1118). Despite such disagreement, industry leaders are seemingly committed to the prospect of partnering tourism development and conservation as a mechanism to forge a sustainable future that simultaneously satisfies human needs and protects ecological systems over the long-term.

Yet, there are concerns with the twin mantras of Galápagos as a "natural laboratory" and "at risk". On one hand, Laso (2020) asserts that "the regions' economic and social development promoted by the narrative of "Galápagos as a natural laboratory" has failed to encourage standards consistent with its status as a UNESCO world heritage site" [10] (p. 146). It is thus critical that Galápagos stakeholders collectively assess the reasons why economic development has fallen short of global sustainability standards and how and to what extent the archipelago should continue to be promoted as a natural laboratory. On the other, the narrative of "Galápagos at risk" overshadows the reality that Galápagos' development model is distorted since it, in fact, accelerates economic leakage and social stratification [5]. This reality resonates with certain critiques of the global development model, wherein Western "development" is declared defunct yet paradoxically promoted as the only way forward [28–30]. Such criticism of the "development" project echoes in the Galápagos context. For example, González et al. (2008) affirm that the archipelago "is shifting to an economic development model that is fundamentally at odds with long-term conservation and sustainability interests" [31] (p. 2). Additionally, Laso (2020) suggests that "Galápagos' development model falls short, as protected areas and the demands from the tourism industry have displaced and overwhelmed traditional practices" such as farming and fishing [10] (p. 146).

The complexities of and apparent paradox with these twin mantras spotlight a perverted reality: Galápagos' tourism industry has steadily bolstered conservation funding and risk assessment over the past half-century while also causing the primary ecological degradation of the archipelago's natural capital over that span. Harm to natural systems is caused directly (e.g., waste disposal, mining raw materials for infrastructure development, energy, and water consumption) and indirectly (e.g., farmland abandonment, the damage invasive species cause to crops and native species) [31–33]. Many critical questions emerge. What risks does tourism development pose for emerging economies globally and in Galápagos? What should the tourism industry's role be in Galápagos to steward long-term environmental and social sustainability? As an introduction, the following two subsections briefly outline critiques of the tourism industry and serve to contextualize Galápagos' food security crises, which are presented in Section 2.5.

### 2.3. Risk and Reward: Global Tourism at Odds

The democratization of travel has enabled an increasing portion of the global community to experience the biosphere's natural wonders, develop cross-cultural competencies, and experience cuisines and events that had only been accessible to some via travel logs, literature, and most recently, social media posts. Prior to the COVID-19 pandemic, the tourism industry accounted for nearly 8.8% of global employment, 5.8% of worldwide exports, and 4.5% of global investments [2]. Tourism has become an increasingly valuable component of foreign trade and is looked to as an opportunity for developing countries to

stabilize export economies. Global tourism is anticipated to increase 3.3% annually from 2018 to 2030—notwithstanding the recessions associated with global financial crises and pandemics [6].

Despite the industry's contributions to the global economy, tourism growth projections raise concern with the industry's ability to account for the increasing pressures it places on the carrying capacity of natural and social systems. For instance, the pre-COVID acceleration of tourism growth was projected to roughly double or nearly triple the industry's environmental impacts by 2050 [2]. To raise awareness and to motivate resilient change, global initiatives (e.g., the United Nations Environment Programme's Rio +20 gathering in 2012) have reported on the multifaceted impacts experienced by host destinations, including but not limited to: strain on water and waste systems, energy-intensive transportation and increasing emissions, damage to terrestrial and marine biodiversity, and the erosion of local cultures and heritage [2].

These issues are especially problematic in developing economies, where the negative impacts of global tourism development are not experienced uniformly and lead to fundamental changes in local ontologies and epistemologies. For example, global tourism practices often produce an asymmetrical sharing of cultures and Westernization in host communities [9,34]. Additionally, residents commonly abandon traditional labor industries to seek higher wages in tourism jobs, which alters informal economies as well as societal norms and values [1,9]. Furthermore, economic leakage is characteristic of and a particular concern for sustainable tourism development, especially in the Global South [2,9]. Economic leakage is understood to accelerate uneven development within host communities, lead to wealth stratification, and thus trigger social discontent among residents regarding how tourism revenue is distributed [5,9,31].

In response to this scenario of underdevelopment, resident discontent and economic leakage to foreign-owned tour operators, several tourism alternatives have emerged: green tourism, sustainable tourism, ecotourism, and regenerative tourism. These progressive tourism models generally strive to develop sustainable futures for future generations by stewarding destinations' natural capital, positioning residents with leadership and ownership stakes, and subverting the extractive tendency of mass tourism operators to sidestep investment in local infrastructure and social services. However, it is important to contest the assumption that mass tourism practices are always inferior to or more harmful than green, sustainable, and regenerative industry alternatives. For example, ecotourism practices are often viewed to dissociate local communities from the lion's share of tourism revenue and to strain a destination's natural capital by requiring greater amounts of resources to access isolated environments [35,36].

The reality, then, is that host destinations experience considerable risk when tourism practices are managed irresponsibly and without the well-being of social and natural systems at the fore. Considering this brief overview of the global tourism project, the next section affirms that Galapagueños are tourism dependent. This dependency means the archipelago's food production industries (e.g., fishing, farming) are increasingly vulnerable to external shocks (i.e., global crises).

### 2.4. Risky Business: Local Tourism at Odds

The onset of Galápagos' cruise boat tourism industry in 1969 triggered a new phase of development [31]. Tourism to and within Galápagos has soared over the past two decades, with land-based tourism growth outpacing boat-based tourism [18,27,37]. Leading up to the COVID-19 pandemic, tourism in Ecuador accounted for the third highest source of non-oil income and 5.1% of the GDP [38]. In 2019, a total of 987,241 total international tourists visited Ecuador [39]. Of those entries, 182,501 visitors entered the GNP [4]. When factoring in the Galápagos' 88,737 domestic tourists in the same year, a total of 271,238 tourists visited the GNP in 2019 [4]. This disproportionate and steadily increasing influx of foreign visitors accelerates tourism profits but risks the archipelago's carrying capacity. The externalities

presented in the previous section are evident in Galápagos, where actors are at odds over resource use, conservation management, and municipal development futures [33].

As a matter of perspective, the total number of 2019 tourist entries is roughly nine times higher than Galápagos' permanent resident population [5]. These pre-COVID tourist visitor statistics indicate that tourism has replaced farming and fishing as Galápagos' main economic driver. Quite simply, Galápagos is currently path-dependent on ecotourism revenue. Pizzitutti et al. (2017) suggest that "nearly 60% of residents are associated with tourism and tourism accounts for nearly 80% of the local economy" [27] (p. 1122).

On one hand, there are perceptions that Galápagos' tourism growth stimulates the island economy by increasing demand for goods and services by tourists and residents alike and thus produces higher wages and increased migration [36]. On the other, ecotourism growth in Galápagos contributes to fundamental socio-ecological and socioeconomic change. Most notably, reduced agricultural and fisheries production is tethered to tourism growth. For instance, many farmers have left their agricultural livelihoods for the higher returns of tourism labor in Galápagos cities such as Puerto Ayora and Puerto Baquerizo Moreno [33,40]. This vocational transition from farming to ecotourism—and the corresponding urbanization and agricultural land abandonment—has accelerated the archipelago's dependency on importing continental foods and products [10,37]. This dependency is troublesome since external economic shocks make island countries vulnerable to food insecurity [41]. Considering Galápagos' socioeconomic landscape, this article now briefly reviews the risks associated with the ways that Galápagos' ecotourism industry undermines the archipelago's food security. An overview of Galápagos' vulnerable food system is provided in the next section to contextualize this study's data and conclusions.

*2.5. Food Systems at Risk*

Several factors have intensified the vulnerability of Galápagos' food systems (e.g., a gradual reduction in agricultural production, restrictions placed on the artisanal fishing sector, an ecotourism boom following Galápagos' recovery from the 2008 global financial crisis, and the COVID-19 pandemic) [42]. Sampedro et al. (2018) suggest that Galápagos' food-supply system is "primarily controlled by population growth, weak local agriculture and imports, and influenced indirectly by the tourism industry," which highlights the "glocal" factors impacting the archipelago's long-term sustainability [8] (p. 2). This scenario means that Galapagueños and local stakeholders face a crossroads when considering post-COVID realities. How and to what extent is it possible to achieve food security amid the pressures that global tourists' steadily increasing consumption poses for food self-sufficiency futures? A brief introduction to Galápagos' food systems on land (agriculture) and then at sea (artisanal fishing) herein is provided as an entry point to analyzing this study's data and, more generally, the future of Galápagos' food security vis à vis ecotourism development.

After Charles Darwin's 1835 arrival to Galápagos, agriculture began in Galápagos in the mid-1800s, as Ecuadorian farmers were attracted to the archipelago's fertile volcanic land [15,40]. Manuel Cobos' colonial station on San Cristobal was responsible for the most aggressive agricultural expansion from 1879 to 1904 [43]. Agricultural production left San Cristobal Island's highlands "denuded of native vegetation" through processes of pastoralism, the cultivation of introduced trees, and the development of export crops (e.g., coffee, sugar cane) [43] (p. 24). The lack of environmental accountability allowed industrial agriculture to develop into one of Galápagos' leading economic engines leading up to the GNP's 1959 establishment. Conservation practitioners labeled agricultural production as antagonistic to conservation development [10]. Since then, the positive advancements in conservationist management plans, (e.g., [44,45]) have prioritized the preservation and restoration of terrestrial ecosystems and thus have placed agricultural practices under a sustainability microscope.

In addition to enhanced conservation management, other shifts have led to reduced agricultural production in recent years. On one hand, there has been accelerated "land

abandonment", which Villa and Segarra (2010) explain is the process wherein farmers abandon agricultural lands at the prospect of higher profits in other sectors—namely tourism [46]. On the other, there has been a reduction in "land clearing" for agriculture. It becomes clear that Galápagos' diminished agricultural production is multifaceted. Yet, it is only a part of Galápagos' food production, both past and present.

The artisanal fishing sector was also instrumental in early communities' pursuit of food self-sufficiency. Pioneering fishermen such as Segundo Asencio developed Santa Cruz Island's "Pelican Bay" fishing wharf beginning in the late 1960s. Their practices developed without restrictions—and organically, as they understood local fish species' habits and how to care for the fish stocks responsibly. The ontologies and epistemologies of these artisanal fishers represent a kind of embedded local knowledge of sustainability. Decades later, Galápagos experienced a mass migration of continental fishers, who arrived in the islands to participate in the sea cucumber boom-and-bust from 1988 to 1992 [47]. Commercial fishing activities, in general, proliferated during the 1990s, to the point of threatening the stability of certain fish stocks [24]. Overfishing was possible since the Galápagos National Park Service's (GNPS) regulatory capacity (e.g., GNP rangers) was ill-prepared to monitor and to prevent the collapsing of fish stocks, which nonetheless has occurred (e.g., Galápagos' endemic sea cucumber species) [48,49]. Consequently, conservationists perceived these fishers' practices as increasingly destructive, attributing fishers with reputations as predators. Meanwhile, pioneering fishers in Puerto Ayora, Santa Cruz Island's port town, sustained local consumption and performed their practices freely. Their efforts earned them reputations as providers, which is a stark contrast to the lingering reputation that fishers hold today as predators and antagonists to conservationist initiatives.

Over the past two decades, the GNPD's tight regulations of fishing calendars and quotas has reshaped artisanal livelihoods, practices, and ways of knowing and interacting in and with the sea. The GNPD authorizes which fishing practices and materials (e.g., hooks, lines, nets) are permissible. Moreover, the GNPD froze the number of fishing boat berths and capped the fisher registry, reducing and then fixing the maximum number of permits available. More than 1000 fishers were active during the 1990s fishing bonanzas. This population has reduced to approximately 300–400 active fishers, who today, supply global tourists' and residents' fish consumption [50]. Several conservationists who were interviewed during fieldwork communicated a hope to terminate the fishing sector altogether, which would end its tenure as "an extractive economic industry" (i.e., overfishing, by-catch), bolster conservation, and preserve marine ecosystems for the ecotourism experience. Other stakeholders, including a GNPD administrator, suggest that fishing can be part of Galápagos' long-term economic infrastructure if developed responsibly and considered as a complement to tourism. He further commented on Galápagos' path dependency on tourism, stating, "If we lose nature, then we lose tourism and our capacity to live in Galápagos. We'd have to migrate elsewhere. It sounds extreme, but it's real. This scenario is happening across the globe." In other words, Galápagos fishers have been re-imagined and conditioned to perform a complementary role in the archipelago's sustainable tourism future.

Amid this eco-political backdrop, artisanal fishers find themselves laboring without the comforts and technologies of commercial fishing vessels and adapting to keep pace with global consumers' demand for pelagic fish, which pushes them to riskier waters and longer periods at sea [5]. Galapagueño fishers voice a range of concerns about how the archipelago's sustainable development has marginalized their marine rights and stewardship of the GMR. Fishers seemingly face the precarious choice between either struggling to maintain their traditional practices or abandoning their industry to seek alternative employment (e.g., marine-related tourism services).

This brief accounting of the diminishing agricultural and artisanal fishing sectors is alarming, especially considering Galapagueños' aspirations for the ecotourism industry to return to pre-COVID numbers of annual tourist entries. It is thus paramount that

Galápagos stakeholders address the confluence of tourism dependency, global pandemics, and food systems to secure the long-term sustainability of the tourism industry and, more importantly, the social well-being of Galápagos residents.

## 3. Materials and Methods

This article draws upon the author's extended qualitative research in Galápagos' marine and terrestrial spaces from 2013–2019. Over that span, the author implemented the Emic Evaluation Approach model, which involves mapping social actors, social discourse analysis, and practice analysis [51]. This fieldwork approach provided opportunities to gain access to and understand Galapagueños' perspectives on socioeconomic futures.

The data collection methods used include archival research, participant observation, multi-sited ethnography, and semi-structured interviews. Archival research served as an initial entry point to understanding the archipelago's histories of development, conservation, and cultural norms and values. Data were collected from various archival sources on San Cristobal Island and Santa Cruz Island, including fishing cooperatives, municipal libraries, and the CDRS' private archive. Additionally, extensive participant observation was conducted across three inhabited islands and the GMR, which engaged a range of social actors, educational institutions, and conservation networks. Examples of participant observation include voyages to various inhabited islands and regions of the GMR aboard an ecotourism catamaran; accompanying artisanal fishers on their journeys across the northern, western, and southern areas of the archipelago; attending inter-sectorial conservation workshops; and joining social actors in their homes, at social gatherings, and in the in-between spaces. This kind of multi-sited fieldwork across "aquapelagic spaces" provided opportunities to gather nuanced perceptions of ways Galapagueños identities are contested and socially re-constructed as well as how sustainability frameworks and eco-political power structures are developed, challenged, and occasionally subverted [5,52–54]. This article also draws upon data from formal and informal interviews collected across multiple islands and vocations (i.e., farmers, tourism operators, restaurant owners, artisanal fishers, tourism industry management, conservationists, educational leaders, municipal planners). Many interviews were conducted in formal settings and were recorded, while others were impromptu at homes or public venues, requiring the author's note taking and journaling immediately thereafter. Additional interviews were conducted virtually from 2020–2021, when travel bans associated with the COVID-19 global pandemic had restricted travel to the archipelago.

## 4. Results

A mapping of interview and fieldwork data and reviewed literature identified several interdependent and complex factors reflecting the Galápagos tourism industry's impact on ecosystems and Galapagueños' well-being. Such factors include but are not limited to: (i) fossil fuel imports and renewable energy production [27], (ii) climate change [41,43,55,56], (iii) invasive species [27,33,40,43,56], (iv) public health issues such as water quality, sanitation standards, and a rise in obesity [19,56–58], (v) GMOs [10,27,57], and (vi) the vulnerability of Galápagos' food systems [8]. Amid the multitude of interrelated factors, three thematic findings are presented to highlight the complexity and vulnerability of Galápagos' food systems vis à vis tourism development. These three themes are that (i) tourist food preferences have fundamentally altered local food systems, (ii) food technologies are gradually modernizing local agricultural production and gaining recognition as a driver of food self-sufficiency development, and (iii) the public sector has made progress in stimulating Galápagos' agricultural food production.

### 4.1. The Impact of Tourism and Conservation on Marine Food Systems

Interview data from various social actors (e.g., restaurant owners, artisanal fishers, fishing wharf laborers, GNPS observers) indicate that the food preferences of global tourists and local conservationist initiatives have significantly altered local cuisine, food system

networks, and fishing livelihoods. Data suggest that a leading factor motivating such change is the fishing sector's general transition from demersal to pelagic fish sales. This change in fish consumption patterns corresponds with ecotourism growth in Galápagos over the course of this study's fieldwork and conservation efforts to protect depleted demersal fisheries, e.g., [49,59].

For perspective, informants revealed that artisanal handline fishing practices in Galápagos have evolved considerably during the past half-century, especially over the past 10–15 years. Generally, the shift in artisanal practice has been one from the handline fishing of a demersal grouper fish, known locally as *bacalao* (commonly referred to as cod and found in coastal areas between 2–73 m), to the "mid-water long line fishing" method in the deep sea of pelagic fish such as tuna and swordfish [60,61]. Puerto Ayora fisher informants recounted that pioneering subsistence fishers in the 1950s predominantly targeted *bacalao* since it was easy to salt and dry or export and did not need refrigeration, which was not available at the time. As one fisherman explained, "fishing systems, arts, technologies, and fishermen have changed dramatically since my youth. Though some fishermen still catch *bacalao* today with line and up to five hooks, new technologies and practices are replacing *bacalao* fishing with a new (mid-water long line fishing) form of tuna fishing at high sea." Numerous mid-water long-line fishers communicated that their changes in practice are market-driven. Local restaurant owners purchase fewer demersal fish to keep pace with global tourists' demand for pelagic fish. However, what accounts for the change in market demand and the shift toward scaling-up a pelagic fishery?

As a starting point, data are presented from fieldwork interviews with restaurant owners adjacent to the "Pelican Bay" fishermen's wharf, which is a hub of activity in Puerto Ayora on Santa Cruz Island. Interview data were collected during a period (2010–2019) when the compound annual growth rate of tourist entries was 51% (173,297 to 271,238) [3–5]. The data indicate that restaurant owners today overwhelmingly prioritize offering pelagic fish dishes (i.e., tuna and whitefish such as wahoo, swordfish) over demersal fish alternatives since serving the former yields higher profits. More specifically, the informants explained several reasons for the market shift toward prioritizing pelagic fish cuisine: (i) pelagic fish cost less per pound, involve less waste when sizing fillets, and are easier and quicker to cook, (ii) foreign tourists recognize pelagic fish names and are often confused by the local demersal fish names (e.g., *bacalao, brujo, camotillo*), (iii) tourists order fish at restaurants at higher rates than locals do, and most importantly, (iv) tourists account for the majority of restaurant sales in this area. These data indicate that the food preferences of tourists have influenced market demand and have incentivized many fishers to switch from demersal to pelagic fishing.

Yet, tourism is not the only driver of change in fish production. The conservation sector also contributes to designing the future of "sustainable" fisheries and, as a byproduct, reconstructing fishers' livelihoods. Tanner et al.'s (2021) study of tourists' preferences and tradeoffs when buying certified fish in Galápagos explains that the recent and growing shift from demersal to pelagic fish sales corresponds with a 2014 initiative developed by the GNPD, the World Wildlife Fund, and Puerto Ayora's fishing cooperative to implement a seafood eco-labeling program [49,62]. Eco-labeling is a certification that is designed to provide the guarantee to the consumer that the product (i.e., yellow-fin tuna) meets high standards such as environmental and social responsibility, quality, and legality [62]. The program also aims to "help local fishers reach a sustainable pelagic fishery, and shift efforts away from depleted coastal ones" by reducing by-catch [49] (p. 4). While the initiative is promoted as a support to fishers' livelihoods (see [49]), my fieldwork data taken while accompanying fishers at high sea reveal that pelagic fishing practices also produce numerous negative social externalities for fishers and their social networks [63]. A key takeaway from Tanner et al.'s (2021) study is recognition that the market shift is a deliberate, institutional effort to "use price signals and consumption preferences to influence the local fisheries production decisions" and to "align market incentives with

conservation," which is achieved by pushing fishers toward yellow-fin tuna fishery and "sustainable" fishing methods [49] (p. 2).

The change in market demand and the modifying of fishing practices have subsequently altered Galapagueños' diets and traditional cuisines. *Bacalao* and other demersal fish are not prominent items on Puerto Ayora restaurant menus. Numerous stakeholders remarked that the tourism industry should be responsive to local food supply (e.g., fish on hand, seasonal produce) and not the inverse. For example, Jimmy Bolaños, former director of Galápagos District Directorate of the Ministry of Agriculture and Ranching (GDDMAR), suggested that restaurant owners and cruise boat operators should take the initiative to adapt their menus to the seasonal fruits and vegetables that are available, which reduces the volume of imported food products that are out of season or that are not locally produced.

Therefore, it is critical to consider that, on one hand, the tourism sector contributes to sustaining fishing livelihoods as the leading local consumer of pelagic fish, yet, on the other, contributes to fundamental changes in food systems as well as local ontologies and epistemologies. This series of developments poses considerable concern for fishing futures in Galápagos, as the industry's small-scale workforce (300–400 active fishers) is ill-prepared to supply residents' and tourists' consumption rates and preferences over the long-term, especially considering that conservationist pressures restrict replacing artisanal methods with industrial technologies.

### 4.2. Modernizing Agricultural Production to Bolster Food Self-Sufficiency

Sustainable food technologies are slowly modernizing local agricultural production and are gaining recognition as a viable component of Galápagos' food self-sufficiency development. The Granja Integral Ochoa, a small-scale farm on Santa Cruz Island, is presented as a case study to contextualize challenges with the projects of modernizing Galápagos' food technologies and incorporating food production as an element of the traditional tourism package.

Granja Integral Ochoa is a two-hectare farm located in the Santa Cruz Island highlands that cultivates various crops (e.g., corn, coffee, plantains, figs). The farm's director, Romer Ochoa, remarked that invasive species and insects had been wiping out high percentages of crop yields. This challenge was a reason for the farm's initial hydroponic enterprise, which began in 2013, when Ochoa viewed a report about the advantages of hydroponic technologies in controlling insects and caring for the environment. His initial attempt at hydroponic production was unsuccessful. Others in the agricultural sector criticized his venture, declaring that hydroponics is not viable in Galápagos. Interview data from administrators at the GDDMAR confirmed the idea that farmers are hesitant to trust innovative ventures. Years later, Ochoa decided to renew the initiative, which required his own financial investment and traveling to Colombia to complete a hydroponics course. State funding and technical training courses on innovative technologies were not available in Galápagos at that time. He dedicated 200 square meters of the Granja Integral Ochoa to the hydroponic production of lettuce (Figure 1). Today, the farm yields roughly 1000 plants every 15 days. Production cycles last 30–45 days and involve 4000 lettuce plants at various stages of development. A bag of hydroponic lettuce today sells for USD 1.50, which is considerably less expensive than imported, traditionally grown lettuce, which sells at USD 3.75.

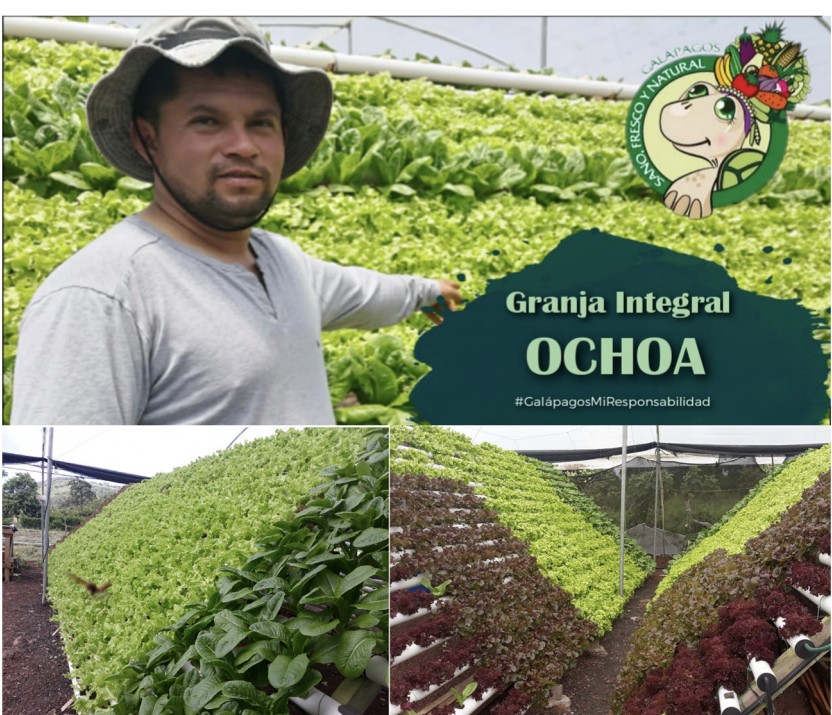

**Figure 1.** Hydroponic lettuce production at Ochoa Integrated Farm on Santa Cruz Island. Photos provided to author by R. Ochoa on 3 August 2021.

Despite the achievement in scaling-up production and the favorable price point of his product, Ochoa laments that tourism operators (i.e., hotels, restaurants, cruise-boats) are largely reluctant to partner with his innovative project and others like it. The Ochoa Integrated Farm case study resonates with data on small-scale farming sales reported in O'Connor Robinson et al.'s (2018) study of the trade-offs associated with pesticide use among Santa Cruz Island farms [40]. The study interviewed 27 farmer households and found that 85% of informants reported selling their crops at local markets. Only one organic farmer had a contract with a tourism operator [40]. Ochoa also noted the difficulty in changing tourism supply networks as well as Galapagueños' consumption norms. Prior to the COVID-19 pandemic, Ochoa's hydroponic lettuce production supplied several small grocery stores, a few restaurants, four cruise boats, and Galapagueños at local markets. That changed when the COVID-19 pandemic upended tourism supply networks, which compromised flows of tourism revenue to and within Galápagos. Data from other Santa Cruz Island farming informants suggest that unemployed tourism laborers journeyed from port towns to highland farms in search of agricultural work during the 2020 tourism freeze. However, this influx of laborers was temporary, as the Galápagos' tourism sector slowly rebounded in early 2021. The lifting of travel restrictions led to the tourism sector's gradual resilience. Tourism revenue flows again motivated agricultural laborers (who worked in tourism pre-COVID) to return to their tourism labor.

The outlook for increased hydroponic lettuce consumption and similar innovative agricultural ventures is uncertain once the archipelago's tourism operations return to pre-COVID levels. However, Ochoa affirmed that other farmers are motivated to explore hydroponic production. A challenge, he stated, is that farmers seldom have the capital to invest in hydroponic technology, and the State and NGOs have not yet invested substantially in progressive agricultural projects. For example, interview data from farmers on Santa Cruz Island and San Cristobal Island inform that public funds seldom reach small-scale farmers and entrepreneurs seeking to invest in environmentally minded agricultural systems (e.g., hydroponics).

The precarity of funding networks and investment flows is a commonly accepted reality in Galápagos. For example, Bocci's (2017) study of migrant farm workers in Galápagos

and their multispecies entanglements concluded that farmers are subject to "a political regime that is absent and contradictory, rather than directly oppressive. Putting aside long-term planning and investments, farmers devise ways to procure a livelihood that is uncertain yet durative" [64] (p. 136). This kind of self-reliance and improvisation resonates with Ochoa's experience. He has pressed onward despite the financial hardship and roadblocks associated with operating a Galápagos farm. The lack of financial sponsorship has left Ochoa to explore agrotourism as an alternative avenue to procuring a livelihood in a changing economic landscape. For instance, he is partnering with a few small-scale tourism operators to offer tours of his farmstead and hydroponic production. The attendant revenue serves as a complement to the farm's traditional production methods.

The trickle of island-hopping tourists via agrotourism also provides Ochoa and other Galápagos farmers opportunities to discuss several sets of relationships: the relationship they and other Galapagueños have with the natural system, the relationship between the tourism sector and the pressures it places on Galápagos' food insecurity, and the ongoing struggle shared by stakeholders to develop the archipelago's food self-sufficiency. The data show that Ochoa and other farmers hope that local stakeholders and residents will increasingly value local food production. Informants suggest that this change in values should be reflected on two fronts. The tourism sector and permanent residents should purchase increasing quantities of local food products instead of imported alternatives. Additionally, the State and NGOs should substantially increase investment in innovative agricultural technologies and other food production projects.

### 4.3. The Public Sector's Commitment to Food Self-Sufficiency

The public sector has defined clear goals and is taking steps to stimulate Galápagos' agricultural food production, which aims to provide positive social and environmental outcomes. For perspective, the GDDMAR (2021) identifies that 98.5% of Galápagos' rural areas designated for human use (i.e., 25,059 hectares) are suitable for agricultural intervention [56]. In addition to animal agriculture in Galápagos (e.g., cattle, pork, chicken, goats), the staple crops include coffee, *yuca* (cassava), plantain, corn, tomato, bell pepper, pineapple, and other tropical fruits (GDDMAR, May 2020). However, only 14,000 hectares of agriculturally zoned land are utilized, yielding 600 tons of food monthly. This yield falls well short of the roughly 1300 tons that Galápagos' 30,000 residents consume monthly [65].

A GDDMAR report (2021) indicates that the total local food production increased by 218% from 2017–2021, yet only accounted for 42% of the province's total demand [56]. There are numerous variables in this relatively low food self-sufficiency metric, especially considering the COVID-19 pandemic. According to Bolaños, the public sector asked Galapagueños to farm during the COVID-19 pandemic to sustain local consumption amid supply network delays triggered by the paralyzed ecotourism industry [66]. To this point, interview data inform that residents with traditionally high-paying and stable jobs (e.g., tourism laborers; GNPS naturalist guides) often found themselves reliant on donated food baskets to withstand income shortfalls amid the COVID-19 tourism shutdown.

Recognizing the vulnerability of Galápagos' food systems, the public sector has established both practical and conceptual goals for agricultural production. These goals seek to mitigate food insecurity in the short term (i.e., COVID-19 related food crises), achieve food self-sufficiency in the long term, and make positive contributions to conservation initiatives. On one hand, the public sector plans to improve food supply networks in the short term. For instance, the GDDMAR strives to reactivate 50% of abandoned farmland [66]. This goal is an essential stopgap in reducing imported foods metrics, which is anticipated to reach 95% by 2037 if resilient actions are not taken [8]. On the other, the public sector looks to socialize the notion that agriculture and conservation are compatible and can be developed simultaneously. For example, Bolaños noted that agriculture, when practiced responsibly, may generate healthy products for the local population and tourism sector while also serving as an ally to conservation [65]. This notion corresponds with Wolford et al.'s (2013) recognition that efforts to unite farmers with NGOs and the GNPD intend to

"align the duality of agricultural and conservation development" [67] (p. 101). Additionally, Carlos Ortega, former Director of the San Cristobal Island GNPD, explained in an interview that agriculture need not be viewed as a threat to conservation since abandoned farms accelerate the settling of invasive plant species such as *mora* (blackberry) and *guayaba* (guava). Furthermore, Ortega noted that farmers have replanted native species (e.g., to restore streams and other spaces to their natural state), which provides birds and other native species protection from invasive predators. In this light, agricultural production, and particularly farmland restoration, is understood to support conservation efforts.

The public sector has taken several actions over the past few years (prior to and continuing through the COVID-19 pandemic) to promote food system development and food self-sufficiency. For example, a sustainable farming workshop on Santa Cruz Island in 2018 informed farmers about "climatically intelligent farming", climate change resiliency, and how to secure credit lines [68]. In June 2019, the GDDMAR supported a USD 172 million proposal titled "Galápagos Compatible with the Climate" that aims to stimulate the archipelago's adaption to and mitigation of climate change [69]. That proposal recommended that Galapagueños adopt a climatically intelligent and inclusive agricultural model as one of several key objectives. The proposal is praised as a step toward showing the world that Galápagos can be food self-sufficient—not only for the 30,000 residents but also for the 276,000 annual tourists [69].

In May 2020, a conglomeration of local stakeholders (e.g., the GDDMAR, the Galápagos Governing Council, NGOs, municipal government) estimated that USD 1.5 million would be required to regenerate 5000 hectares as productive farmland [65]. The funding endeavors to modernize agricultural equipment, generate communal and urban gardens, and develop educational infrastructure among other outcomes, a project that is underway. For example, the GDDMAR (2021) reported that there was a 181.5% increase in irrigation system coverage from 2017–2021 [56]. However, interview data suggest that investments in agricultural technologies commonly seek to scale-up traditional infrastructure and practices (e.g., standard irrigation systems) and have not yet invested heavily in hydroponics and the use of drones to inform planting decisions and manage invasive species. Nonetheless, the USD 1.5 million financial injection into the agricultural sector is projected to produce an additional 200 tons of food per month and thus increase the total monthly yield of locally produced food to 800 tons [65].

These data collectively highlight the vulnerability of Galápagos' food systems and spotlight several issues with marine and terrestrial food production and its relationship with the pressures from tourism development. The following section discusses the findings by presenting several key themes that emerge from the data: the tourism sector's role in accelerating food insecurity, deficient investment in sustainable food systems, Galapagueños' right to food sovereignty, and the implications of Galápagos' sustainable tourism project for other Pacific destinations and for strengthening partnerships that seek the compatibility of ecotourism and food systems.

## 5. Discussion

The journey toward self-sufficiency in Galápagos is gaining momentum. Yet, residents still face considerable challenges in developing resilient food systems, especially considering how the COVID-19 pandemic has exposed their vulnerability to global crises and the resulting supply chain disruptions. Several discussion points are presented herein to speak to the relationship between Galápagos' tourism model and the archipelago's vulnerability to food insecurity.

First, the study's findings indicate that Galápagos' current ecotourism project exacerbates the vulnerability of Galapagueño food systems. The tourism sector does little to address the externalities and risks that tourism practices pose for Galapagueños' well-being. Not only do ecotourism practices minimize local communities' histories, cultures, and ecological values in the touristic experience, but the practices also enable exogenous tourism owners to extract profits while conveniently passing the responsibility to develop

municipal infrastructure (i.e., food, energy, and waste systems) to the public sector. This economic structure entrenches a development model, e.g., [27,37] that González et al. (2008) characterize as "continentalized or exogenous" and "poorly adjusted to the fragility, uniqueness, and particularities of the archipelago" [31] (p. 11). In this light, Galápagos' tourism model, in fact, (i) accelerates food insecurity instead of using the touristic experience as an opportunity to spotlight human systems' vulnerability to exogenous markets and, more importantly and (ii) sidesteps significant investment in the long-term modernization of food systems and thus the sustainability of Galápagos' food basket. Global crises will magnify these issues as annual tourism numbers rebound to, and likely surpass, pre-COVID levels.

Second, the findings call attention to the apparent dissonance between Galapagueños' current food insecurity and their constitutional right to food sovereignty. The collected data indicate that tourists' food preferences have fundamentally altered local food systems, such as changes in fish yields and restaurant offerings. Food sovereignty in Galápagos is thus a matter of supporting Galapagueños' right to develop and govern productive food systems that prioritize the needs of future Galapagueños and not simply cater to tourists' culinary tastes [70–72]. However, the right to food sovereignty transcends idealism. The 2008 Ecuadorian constitution incorporated food sovereignty "as an entitlement for the attainment of *sumak kawsay*," a Kichwa indigenous habitus that embodies "the values of social justice, inclusion and equality" [71] (p. 225). In this way, food sovereignty in Galápagos involves the state's duty to achieve food self-sufficiency that includes healthy and culturally appropriate food [71]. Therefore, it is important for local actors to dialogue as to how and to what extent the global ecotourism project in Galápagos compromises the residents' constitutional right to food sovereignty and what actionable steps can be taken to remedy this situation.

Third, the findings contribute to meaningful discourse on the fate of sustainable island tourism and food security in the Pacific region. For example, the Galápagos case study aligns with McGregor et al.'s (2009) recognition that the level of food self-sufficiency is a critical measure of food security in "small, vulnerable and sometimes unstable Pacific Island economies" [41] (p. 29). On one hand, the Galápagos context benefits when applying lessons learned from other studies in the Pacific (and beyond), especially discussions on ways to develop localized food production to stabilize tourism-based economies amidst global crises. To this point, Galápagos stakeholders should consider drawing inspiration from Henderson's (2018) study of how a culturally relevant and staple crop in Hawai'i called *'ulu* (breadfruit) has potential to be developed as a reliable food source to strengthen disaster preparedness, food security, and community-based economic development [73]. Henderson's (2018) study resonates with fieldwork data that calls attention to how Galapagueños' food preferences are often incompatible with Galápagos' soils and growing cycles as well as the archipelago's food self-sufficiency goals. For example, Bolaños remarked that while rice is difficult to grow in Galápagos' terrain, it remains a seemingly unshakeable staple of Galapagueños' traditional diet. He remarked that while Galapagueños are unlikely to give up eating rice, even a small-scale shift to carbohydrate substitutes (e.g., *yuca*/cassava, potato, sweet potato) would spark growth in local supply. Doing so would also assist with disaster preparedness, as locally produced and accessible foods would provide a stopgap when global pandemics disrupt supply chain networks.

On the other, the Galápagos context offers valuable lessons to other Pacific archipelagos that have committed to tourism as an escape from "underdevelopment", which Frank (2014) explains is a state produced by the development of (global) capitalism [74]. At a conceptual level, Bocci's (2017) study among Galápagos migrant farming communities models the practice of replacing *crisis* language with conversations of *emerging assemblages* [64]. A focus on assemblages in Galápagos reinforces the idea that the project to build just and sustainable (food/tourism) futures should be collaborative and should transcend eco-political boundaries—both the imagined boundaries that separate actors and institutions within

Galápagos as well as the eco-political boundaries that isolate the GNP and GMR from the exterior.

Fourth, this study reinforces the belief that developing resilient food systems in Galápagos requires strengthening, and perhaps re-imagining, the existing inter-institutional and inter-sectoral relationships. By conceptualizing the project of building Galápagos' socioeconomic and ecological futures as relational, opportunities arise to embrace the complexity of the archipelago's ecological and socioeconomic issues. In this way, social actors from across sectors are positioned to redesign the stewardship of Galápagos' natural and human systems. This process is likely to stir up concern over the tourism industry's and the public sectors' deficient investment in modernizing and expanding food self-sufficiency. However, such investment commitments require a dedication to partnerships that transcend Galápagos' management and economic silos. Collaboration is indeed recognized globally as a sustainability competency, which is paramount to the efficacy of strategic, anticipatory, and systems thinking [75,76]. Collaborative partnerships are thus key to cultivating the belief that conservation can be compatible with developing food systems and ecotourism if done responsibly and with the well-being of natural and human systems in mind. A practical step toward understanding and developing partnerships among stakeholders and interest holders in Galápagos' is to conduct a social constructivist analysis of food security vis à vis tourism. This action draws inspiration from Zapata's (2005) study of how various social actors attribute power, legitimacy, and interest to users of the GMR [77].

In conclusion, while the COVID-19 pandemic has magnified Galápagos' food crises, the tourism shutdown has also given local actors cause to fundamentally assess food networks and consumption habits. Looking ahead, the Galápagos tourism industry's resilience is seemingly inevitable. Annual tourist entry numbers are soon likely to equal if not surpass pre-COVID tourism rates. However, is the goal simply to restore tourism revenue flows at the cost of significant social and ecological externalities? How and to what extent might a regenerative tourism model mitigate issues of food sovereignty and economic leakage? If action is not taken in the short term, the continued acceleration of exogenous food imports may push Galapagueños into a permanent state of food insecurity, and one that is not resilient to global crises. Therefore, the current "building back better" process should consider food systems as the linchpin to the tourism industry's long-term success and, more importantly, the future well-being of Galápagos' natural and human systems.

**Funding:** This research received no external funding.

**Institutional Review Board Statement:** The initial data collection was conducted according to the guidelines of the Declaration of Helsinki, and approved by the Board of Ethics at the University of Cape Town in 2013.

**Informed Consent Statement:** Informed consent was obtained from all subjects involved in the study. Written informed consent has been obtained from the informant(s) to publish this paper.

**Data Availability Statement:** Not applicable.

**Conflicts of Interest:** The author declares no conflict of interest.

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
