# Peer review of "The Crossroads of Ecotourism Dependency, Food Security and a Global Pandemic in Galápagos, Ecuador"

_sustainability, doi:10.3390/su132313094_

Round 1

Reviewer 1 Report

The article contains relevant content and fits into the Special Edition. However, it requires shortening this content, a clearer structure and showing the most important conclusions from the research in the form of illustrations, for example, figures or diagrams. In its current form, the content is too long and tedious. However, it is worth refining because of its importance.

Author Response

This feedback is much appreciated and has helped to resolve several issues in the writing, including but not limited to the following:

1) The reviewer commented that the text is too long/tedious and needs to be shortened. To this point, the manuscript has been shortened by removing numerous sentences/sections that were not essential to the article’s main foci.

2) The reviewer commented the article's structure be developed more clearly. Multiple revisions have been made to the Results and Discussion section to provide clearer structure. In particular, sub-headings have been added to enhance the structure and present the results clearly. The clarity of text and structure has also been improved by shortening the text (#1 above).

3) The reviewer suggested that data may be presented in the form of figures/diagrams. In anthropological writing, it is not common to present qualitative data in the form of figures and diagrams (hence the article’s current format). That said, the reviewer has offered valuable guidance and I will incorporate figures/diagrams in my future writing.

I appreciate the reviewer's comment that this article is worth reading, offers relevant content and fits nicely into the Special Edition.

Reviewer 2 Report

              478 / 5000  

Wyniki tłumaczenia

It is an interesting and valuable scientific study. For the sake of greater clarity of the text, it would be useful to include tables presenting dynamic data in the adopted research period and graphical elements in the form of ... The quantitative and valuable information presented in the text is mainly static or it only presents the dynamics of the analyzed phenomena in a limited horizon. 

Author Response

I am encouraged to read that the reviewer finds this article to be interesting and a valuable scientific study.

While the "Review Report" shows that all sections fall under the 'Can be Improved' category, I have nonetheless made several revisions to improve the manuscript, including but not limited to the following:

1) The reviewer remarked that there can be greater clarity of text. I have removed numerous references/sentences (see sections 2-4) that are not essential to the article’s argument/foci. Also, I have added sub-headings to improve the clarity of text and flow between the key findings presented.

2) The reviewer's comment on static/dynamic data is well-received. I have revised the Methods section to avoid the previous confusion about the scope of data. The manuscript now better indicates the dynamic nature of the data. Also, it is uncommon to present data in detailed tables/graphs in anthropological writing. Nonetheless, the reviewer's feedback is valuable. In the future, I will work to incorporate tables/figures/illustrations to better present the data.

The reviewer's feedback is well-received and appreciated.

Round 2

Reviewer 1 Report

Congratulations, the article is good now!

Author Response

Thank you for the feedback in the first report.

I'm happy to have applied the changes and it is good to know the article is now ready.

Reviewer 2 Report

The Author only partially complied with the comments made in the review. It would be useful to introduce quantitative material presented in dynamic terms into the text, which would present changes in the analyzed phenomena. 

Author Response

The PhD research and subsequent fieldwork leading to this article’s production was qualitative. The research design was guided and reviewed by the Department of Social Anthropology at my university of study. The discipline of social anthropology developed and continues to use participant-observation as its primary mode of data collection, which is a qualitative method used to understand emic perspectives/ontologies/epistemologies over an extended period. Anthropological writing commonly uses ‘thick description’ (championed by anthropologist Clifford Geertz) to present the qualitative data in rich/dynamic ways – such as vignettes to illustrate the ‘softer’ side of lived realities. Other anthropological methods include, but are not limited to, life histories, semi-structured interviews, focus groups, archival research, and even surveys. However, surveys are an uncommon data collection method. The doctoral review board that reviewed/guided this project requested that participant-observation be the primary data collection tool in lieu of surveys. In this light, quantitative data were not included from the onset of the research design for this project.

Considerable structural changes have been made to the dissertation structure to meet the “Sustainability” journal’s organizational structure and presentation of data (i.e. Introduction, Materials & Methods, Results, Discussion). For instance, ‘thick description ’is not used in this article write-up to better present the Results in simplistic terms. Specifically, data on the Ochoa family farm (section 4.2 in the revised manuscript) does not include extensive qualitative data (e.g. an overview of the farm setting/fieldsite, analysis of the farm’s division of labor and the attendant norms/values/beliefs, qualitative data to inform the reader of change in farmers’ epistemologies over time), which would commonly be included in an anthropological ethnography.

I understand that the reviewer is requesting that the paper include quantitative data in the forms of tables/graphs to illustrate the analyzed phenomena in dynamic terms. This is valuable feedback and, as stated previously, I sincerely plan to incorporate this guidance (i.e. the inclusion of quantitative data/tables/graphs) into my future research. In other words, this review process has helped me to see that there is great opportunity to provide ‘mixed methods’ research (i.e. both quantitative and qualitative data) when speaking to analyzed phenomena. More specifically, I plan to incorporate ‘mixed methods’ research design in my future research – and as part of my forthcoming IRB applications.

I hope that this clarification provides insight into the qualitative/anthropological research design for this study, which perhaps was unclear in my previous submission. I hope that this clarification is to the satisfaction of the reviewer. If that is not the case, then I am willing to work closely with the reviewer to identify specific aspects that the qualitative data that might be represented/converted into tables/graphs – and where that data may be inserted into the revised manuscript. Specific guidance would be beneficial to understand the precise process of achieving this task.

Round 3

Reviewer 2 Report

Thank you to the Author for the explanations.

I accept the article for publication in its current form.